# Easy Cell Detachment and Spheroid Formation of Induced Pluripotent Stem Cells Using Two-Dimensional Colloidal Arrays

Goshi Kuno [1,2,*] and Akikazu Matsumoto [2,3]

1    Department of Functional Polymer Research Laboratory, Tosoh Corporation, 1-8 Kasumi, Yokkaichi 510-8540, Mie, Japan
2    Department of Applied Chemistry, Graduate School of Engineering, Osaka Prefecture University, 1-1 Gakuencho, Sakai 599-8531, Osaka, Japan
3    Department of Applied Chemistry, Graduate School of Engineering, Osaka Metropolitan University, 1-1 Gakuencho, Sakai 599-8531, Osaka, Japan
*    Correspondence: goshi-kuno-ye@tosoh.co.jp; Tel.: +81-59-364-5798; Fax: +81-59-364-5489

**Abstract:** Induced pluripotent stem cells (iPSCs) may develop into any form of cell and are being intensively investigated. The influence on iPSCs of nanostructures generated using two-dimensional colloidal arrays was examined in this study. Colloidal arrays were formed using the following procedure. First, core–shell colloids were adsorbed onto a glass substrate using a layer-by-layer method. Second, the colloids were immobilized via thermal fusion. Third, the surface of the colloids was modified by plasma treatment. By adjusting the number density of colloids, cultured iPSCs were easily detached from the substrate without manual cell scraping. In addition to planar culture, cell aggregation of iPSCs attached to the substrate was achieved by combining hydrophilic surface patterning on the colloidal array. Multilayered cell aggregates with approximately four layers were able be cultured. These findings imply that colloidal arrays might be an effective tool for controlling the strength of cell adhesion.

**Keywords:** colloidal array; self-assembly; layer-by-layer; patterning; induced pluripotent stem cell; spheroid





## 1. Introduction

Induced pluripotent stem cells (iPSCs) can differentiate into all types of cells [1] and are widely investigated in regenerative medicine [2,3], pathology modeling [4,5], and drug screening [6]. iPSCs are usually cultivated on dishes with flat surfaces. The surface structure has a substantial influence on cells, and research into the effect of surface roughness on iPSCs is crucial. The linearly aligned concave structure has been reported as a typical application of surface structures to control cell elongation [7]. It has also been discovered that surface features of a size similar to that of the cells allow the cells to elongate on the curved surface, thereby generating focal adhesions differently than on a flat surface [8]. Furthermore, nano-to-submicron structures much smaller than cells affect migration and proliferation [9,10]. Consequently, using surface features in iPSC culture is predicted to allow the control of cell shape, which is not achievable with flat surfaces.

Nanoimprinting and self-assembly are commonly employed for producing nanostructures. Surface structures may be produced using molds in nanoimprinting [7], which increases the difficulty and productivity of their use as the size of structures decreases from the submicron to the nanoscale. Conversely, bottom-up approaches, such as colloidal arrays, have the potential to form nanostructures with high productivity.

The surface roughness of colloidal arrays may be adjusted by colloid size, allowing for easy control of its influence on cell adhesion and proliferation [11]. Another example of the application of colloidal arrays to cell culture has been reported for arrays of

gold colloids with immobilized arginyl-glycyl-aspartic acid peptides [12], tubes [13–15], and poly(N-isopropylacrylamide) microgels [16]. A method for forming hole structures using colloids as templates has also been identified [17]. Other reported self-assembly approaches include coating surfaces with tiny bubbles [18] and employing the self-assembly of alumina oxide [19].

In addition to managing cell properties via surface structure, attempts have been made to regulate cell alignment by surface hydrophilicity and hydrophobicity [20–22]. For example, the patterning of cell-nonadhesive substances, such as polyethylene glycol [23] and polyvinyl alcohol [24], has been reported. The patterning of cell sticky molecules, such as integrins [25], extracellular matrices [26], and peptides [27], has also been observed. Recently, it has been reported that patterning can be used to localize cells and form spheroids that adhere to the substrate [28–31]. Spheroid formation mimics the embryoid body and promotes the differentiation of iPSCs. Spheroids are typically generated by floating culture on 96-well plates with no adherent cells. An automated spheroid culture system can be constructed using adherent spheroids prepared by patterning, which are easy to handle. However, in the patterned culture procedure, obtaining spheroids after incubation is difficult.

This report presents the control of iPSC adhesion using two-dimensional colloidal arrays. iPSCs cultured on colloidal arrays were found to be easily detached from substrates. We also focused on the notion that the surface structure can facilitate cell aggregation [32] and investigated the generation of uniform-sized spheroids by patterning colloidal arrays or a hydrophilic polymer. These findings have several applications, ranging from controlling the strength of cell adhesion to the formation of uniform spheroids.

## 2. Materials and Methods

### 2.1. Materials

Styrene (St, 98%), sodium p-styrenesulfonate (NaSS), and potassium persulfate (KPS, 95%) were purchased from Wako Pure Chemical Industries (Osaka, Japan). The silane coupling agent 3-methacryloxypropyltrimethoxysilane (MPTMS, 95%) was obtained from Shinetsu Chemical (Tokyo, Japan). Poly(diallyldimethylammonium chloride) (PDDA, Mw = 400,000–500,000, 20% solution) was purchased from Sigma-Aldrich Corporation. Toyo Gosei (Tokyo, Japan) supplied the cell-nonadhesive polymer (product name: BIOSURFINE-MRH-AWP, 6 wt% solution). Silica colloids with an average diameter of 450 nm (product name: Snowtex MP-4540, 40 wt% dispersion) were obtained from Nissan Chemical Industries (Tokyo, Japan). Matsunami Glass supplied the glass substrates (Osaka, Japan). Nitto Denko provided surface protection film (product name: E-MASK) (Osaka, Japan). Ethylenediaminetetraacetic acid (EDTA, 0.5 M, pH 8.0), phosphate buffer solution (PBS)(-), StemFit AK02N, and CultureSure Y-27632 were purchased from Wako Pure Chemical Industries. The iMatrix-511 solution was purchased from Nippi (Tokyo, Japan). TrypLE Express Enzyme 1× was purchased from Thermo Fisher Scientific K.K. (Tokyo, Japan). Alexa Fluor 488 antihuman stage-specific embryonic antigen (SSEA)-4 antibody (cat. no.: 330412) was purchased from BioLegend (San Diego, CA, USA). All reagents were used without further purification.

### 2.2. Synthesis of Core–Shell Colloids

Ion-exchanged water (18 g) and 40 wt% aqueous dispersion (0.5 g) of an anionic silica colloid (450 nm in diameter) were mixed in a glass flask. Using nitrogen bubbling, the silica colloid dispersion was degassed for 30 min. Then, MPTMS (0.02 g) was added, and the mixture was agitated for 30 min at 300 rpm using a magnetic stirrer under a nitrogen environment. NaSS (0.08 g) and St (0.8 g) were mixed with ion-exchanged water (10 g), added to the colloid dispersion. Thereafter, the mixture was stirred at 300 rpm for 2 h. Next, the mixture was heated to 65 °C, and KPS (0.02 g) dissolved in ion-exchanged water (10 g) was added. The reaction was performed under a nitrogen environment at 65 °C for 3 h while being agitated at 300 rpm. After the reaction, all colloids were precipitated

by centrifugation at 4000 rpm for 10 min. Finally, the supernatant was removed through decantation. The colloids were redispersed into the ion-exchanged water. To yield pure $SiO_2$–PSt core–shell colloids, the procedure was performed twice.

### 2.3. Coatings of the Core–Shell Colloids Using the Layer-by-Layer Method

The synthesized core–shell colloids were coated using the layer-by-layer method. First, a glass substrate was submerged for 10 s in a 1 mg/mL PDDA aqueous solution before being rinsed with ion-exchanged water and blown dry. The 2 wt% core–shell colloid dispersion was set on the PDDA-coated glass substrate. The substrate was rinsed with ion-exchanged water after 10 s and then submerged in boiling water for 5 min. Next, the substrate was cooled to room temperature and dried with blowing air. Finally, the surface was treated for 30 s using a vacuum plasma system (PIB-20 from Vacuum Device Corporation) at a gas pressure of 20 Pa and current of 20 mA. Before patterning the colloids, the glass substrate was laminated with a surface protection sheet with 200 m diameter pores, and the same technique as stated before was followed.

### 2.4. Patterning of the Cell-Nonadhesive Polymer

Ion-exchanged water (1.04 g) and methanol (4.16 g) were added to a 6 wt% solution of BIOSURFINE-MRH-AWP. This dispersion was spin-coated onto a substrate with core–shell colloids at 2000 rpm and dried at 60 °C for 2 min. Thereafter, the coated film was cured for 2 min under the UV irradiation of a mercury lamp. On this coating, a stainless steel plate with 200 μm diameter holes was mounted, and vacuum plasma treatment was performed for 1 min at a gas pressure of 20 Pa and current of 20 mA.

### 2.5. Culture of iPSCs

Prior to cell culture, culture plates were sterilized by UV irradiation for 30 min. StemFit AK02N (0.2 mL/cm$^2$) with Y-27632 (10 μM) and iMatrix-511 solution (2.5 μL/mL) was added to a culture plate. iPSCs (cell line 201B7, 260 cells/cm$^2$) were seeded on the plate, and cells were cultivated at 37 °C in a 5% $CO_2$ environment. After 24 h from cell seeding, the culture medium was replaced to StemFit AK02N without Y-27632 or iMatrix-511 solution. After 7 days of culture, the medium was removed, and the cells were rinsed with PBS(-). To collect the cells, 1 mL of TrypLE-EDTA (1:1 volume mixture of TrypLE Express Enzyme $1\times$ and 0.5 mM EDTA solution) was added, and the cells were incubated at 37 °C for 2 min. After removing the TrypLE-EDTA solution by aspiration with a pipette, the cells were washed with PBS(-). StemFit AK02N with Y-27632 was gradually introduced.

### 2.6. Fluorescent Staining of iPSCs

The cultured iPSCs were centrifuged and separated from the medium. The cells were treated with 4% paraformaldehyde, followed by repeated centrifugation and washing with PBS(-). A solution of PBS(-) that contained bovine serum albumin (1 wt%), EDTA (0.25 mM), and Alexa Fluor 488 antihuman SSEA-4 antibody (0.25 μg/mL) was added. The fluorescent staining reaction was performed for 3 days at room temperature. To remove free SSEA-4 antibody, the cells were washed with PBS(-) and centrifuged repeatedly.

### 2.7. Characterization

Surface structures were measured using an atomic force microscope (AFM; SPM-9600 from Shimadzu Corporation). The cantilever used was BL-AC40TS-C2. Additionally, a laser microscope was used to study the patterned colloids (VK-X200 from Keyence Corporation, Tokyo, Japan), and the iPSCs were observed under a fluorescence microscope (BZ-X800 from Keyence Corporation, Tokyo, Japan).

### 2.8. Ethical Statements

The 201B7 cell line was obtained from Center for iPS Cell Research and Application, Kyoto University under a contract with iPS Academia Japan, Inc., Kyoto, Japan.

## 3. Results and Discussion

### 3.1. Effects of Colloid Density on iPSC Proliferation and Adhesion

Walter et al. studied the effect of close-packed colloidal arrays on cell adhesion using colloids with diameters of 80–1065 nm [11]. They reported that cell activity was drastically reduced around the 357 nm particle size. Particle size of 200 nm or less had a minor effect on cells compared to smooth surfaces, whereas cell activity was worse at a particle size of 1065 nm. Therefore, particles of 450 nm diameter were used in this study, which had effects on cell adhesion without significantly compromising cell activity. Generally, iPSC culture employs polystyrene (PSt) dishes treated with corona or plasma, a process known as tissue culture (TC) [33]. TC treatment boosts cell proliferation by adding functional groups to PSt, such as carboxyl and amino groups. Therefore, we employed the following procedure for forming colloidal arrays. Figure 1a illustrates these procedures. First, core–shell colloids were synthesized by coating the surface of silica colloidal particles with PSt. The conventional seed polymerization method was employed for the colloid surface coatings. Second, the colloids were coated on the substrate using the layer-by-layer method, which electrostatically adsorbs the particles on the substrate. The colloid density on the substrate was adjusted by changing the concentration of colloid dispersion. Thereafter, the surface of the core–shell colloids was thermally fused by heating to immobilize them firmly to the substrate. Finally, surface modification of the colloid-coated substrate was performed by plasma treatment to improve cell adhesion to the PSt surface. Figure 1b shows an AFM image of the colloid-coated substrate prepared via these procedures. The colloids were randomly dispersed on the substrate.

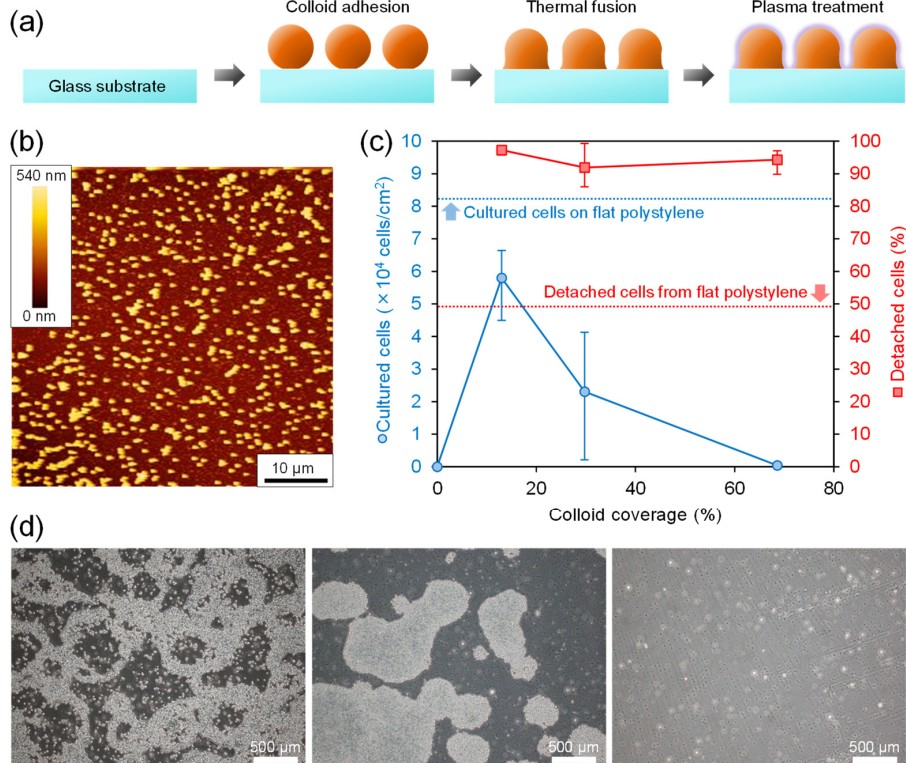

**Figure 1.** (**a**) Schematic showing the process of colloidal array formation and surface treatment. (**b**) Atomic force microscopy (AFM) image of a colloidal array. (**c**) Graph showing the proliferative and detachment properties of induced pluripotent stem cells (iPSCs). Colloid coverage was determined from the AFM image. The horizontal lines are the result of using a common PSt petri dish. (**d**) Phase-contrast microscope images of iPSCs cultured and detached from a typical PSt dish (left image), cultured (middle image), and detached (right image) from the colloidal array. The colloidal array was prepared using 2 wt% core–shell colloid dispersion.

The effects of colloid number density on the proliferation and adhesion of iPSCs were investigated. Figure 1c shows the relationship between colloidal coverage on the surface and the number of proliferated iPSCs 7 days after culture. Only viable cells were counted as proliferated iPSCs by staining the cells with trypan blue. The horizontal lines in the graph are the result of using a general PSt petri dish. The colloidal coverage was calculated by binarizing the AFM images of the substrate surface and determining the percentage of the area where particles were present (Figure S1, Supplementary Information). As coverage increased, the proliferation of iPSCs decreased. The detachment of iPSCs cultured on these samples was examined (Figure 1c,d). Typically, cultured iPSCs are detached by treating them with enzymes, such as trypsin, and then vigorously scratching the surface of the dish with a cell scraper. Manual scraping is an obstacle in the mass culture process of iPSCs. It was discovered in these samples with colloidal arrays that practically all iPSCs could be collected by gently pouring the culture media after trypsin treatment. These results indicate that although the presence of colloidal arrays reduces the proliferative potential of iPSCs, cell proliferation is possible while maintaining a weak adhesive state if particle density is controlled.

In our previous study, we reported the selective detachment of iPSCs from cocultured iPSC-derived differentiated cells through the formation of nanostructures using a thermoresponsive polymer [34]. Because iPSCs detached easily from colloidal arrays, coculture of iPSCs and iPSC-derived mesoderm cells was performed to confirm the selective detachment. Culture conditions are shown in Figure 2a. After iPSCs were cultured to form colonies, mesoderm cells were seeded. Figure 2b shows the flow cytometry results of collected iPSCs from cocultured cells on a colloidal array. Fluorescent staining was performed using iPSC-specific SSEA-4 as a marker. The collected cells contained a large number of mesoderm cells. The results of microscopic observation during cell detachment also confirmed that almost all cultured cells were detached. These results suggest that colloidal arrays also reduced the adhesion strength of mesoderm cells, resulting in no selective detachment of iPSCs. Furthermore, mesenchymal stem cells, HepG2 cells, umbilical vein endothelial cells, gingival epithelial progenitor cells, and epidermal keratinocytes were tested for growth on colloidal arrays and were found to detach just as easily as iPSCs. From these results, we discovered that easy cell detachment using colloidal arrays can be applied to various cells.

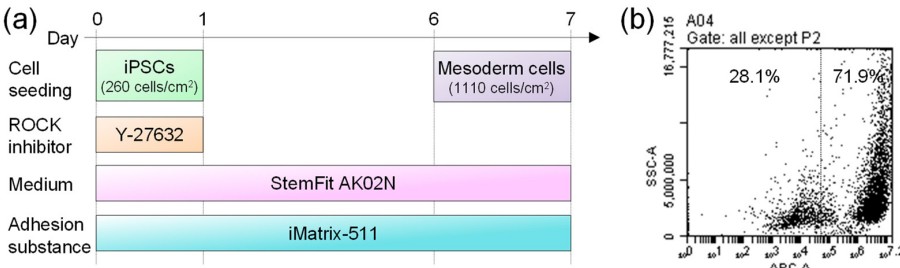

**Figure 2.** (**a**) Flow diagram of the coculture of induced pluripotent stem cells (iPSCs) and mesoderm cells. (**b**) Flow cytometry findings of iPSCs cocultured with mesoderm cells and removed from colloidal arrays. iPSCs were stained using fluorescently labeled antihuman SSEA-4 antibodies.

### 3.2. Spheroid Formation Using Patterning of a Colloidal Array or Hydrophilic Polymer

Using the microscopic pattern of colloidal arrays, a spheroid culture system where cells locally aggregate and was easily detached after incubation was analyzed. Figure 3a shows the patterning procedure for colloidal arrays. Although various methods of colloidal lithography that utilize the self-assembly of colloids have been reported [35], we employed a simple and reproducible method for forming arbitrary patterns using a surface protection film with holes formed by laser processing as a mask. The colloids were arranged in a circular pattern (Figure 3b,c). Because iPSCs were nonadhesive to the bare glass substrate, cells were discovered to attach solely to the colloid region (Figure 3d). This approach is the

same as that used in previous studies to pattern cell affinity materials on flat surfaces [24–27], and it seems that cell adhesion is controllable by changing the colloid surface component. In addition, surface roughness can be changed by changing the colloid size, allowing for more sophisticated surface design for different purposes. However, this method is unsuitable for mass-producing culture substrates because the costly laser-processed surface protection film is disposable.

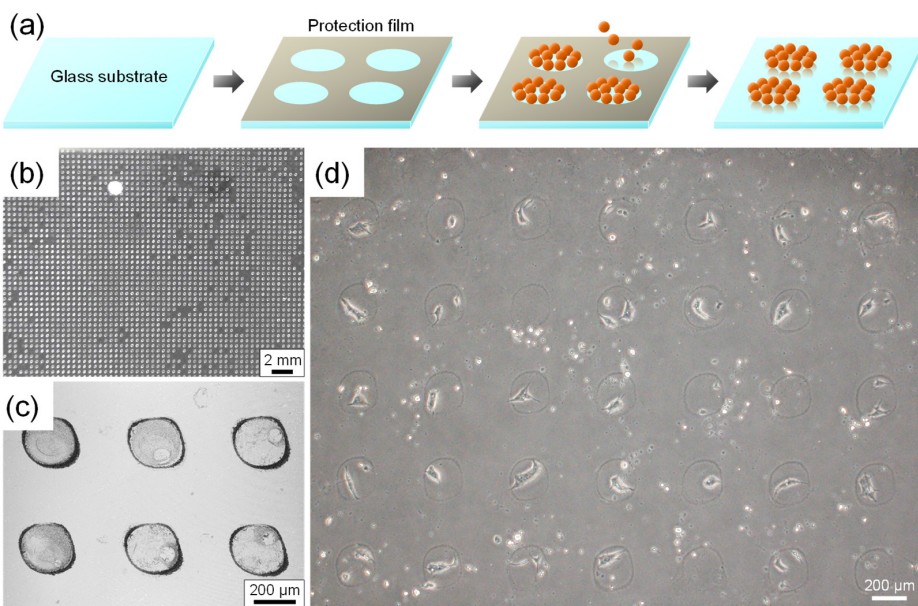

**Figure 3.** (**a**) Schematic showing the patterning process of core–shell colloids. (**b**) Digital camera and (**c**) laser microscope image of the colloidal patterning surface. (**d**) Phase-contrast microscopy images of induced pluripotent stem cells cultured on a colloid pattern.

Patterning of a cell-nonadhesive hydrophilic polymer on colloids covered across the entire surface was investigated as an alternate patterning strategy. Figure 4a shows the procedure. Because patterning is accomplished via cost-effective plasma treatment, this technology is appropriate for mass manufacturing. Figure 4b depicts the results of culturing iPSCs on this substrate, in which cells partially aggregate. Cultured aggregates were easily detached by gentle pipetting or a short enzymatic treatment (Figure 4c).

In contrast, the aggregates stuck to the flat surface were securely attached and required lengthy enzymatic treatment, which resulted in spheroids collapsing throughout the treatment. There have been several reports of the formation of adhesive spheroids by patterning surface polarity [28,30,31]; however, none of them has examined the detachment of spheroids after incubation. Although this reported method has an advantage in that a large number of spheroids can be cultured, it is also important to develop techniques for retrieving cultured spheroids in order to use them for drug discovery testing or transplantation. Kim et al. reported spheroid detachment using thermoresponse [29], which is interesting as spheroids can be detached uniformly by applying external stimuli. One disadvantage of this approach is that some cells may be damaged upon cooling. In contrast, our method forms weakly adherent spheroids, which may reduce damage to cells upon detachment. Based on the above results, it was discovered that the combination of colloidal arrays and patterning made spheroid culture easier to handle.

To analyze the condition of cells cultured on the patterning surface in detail, cells were stained with iPSC-specific SSEA-4. Fluorescence microscope measurements confirmed that the cultured aggregates expressed SSEA-4 and maintained the undifferentiated state of iPSCs (Figure 5a). Scans were made in the z-axis direction at 1 μm intervals to measure the thickness of the cultured cells. The aggregates had a thickness of $61 \pm 7$ μm (Figure 5b,c). Some cells were monolayered. In this case, the thickness was $15 \pm 2$ μm (Figure 5d,e).

These results suggest that the aggregates are stacked with cells in approximately 4 layers. Monolayer cells had weak SSEA-4 expression and differentiated into other cells. As typified by mesenchymal cells, some cells stop proliferating upon contact inhibition. Therefore, it is thought that multilayering did not occur in this area.

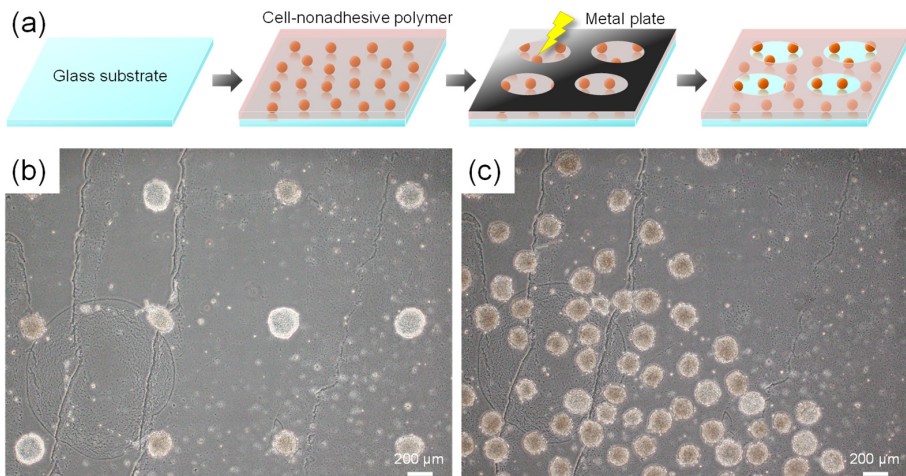

**Figure 4.** (**a**) Schematic showing the patterning process of cell-nonadhesive polymer. (**b**) Phase-contrast microscope images of induced pluripotent stem cells cultured on a cell-nonadhesive hydrophilic polymer patterned substrate for 6 days and (**c**) after detachment by gentle pipetting.

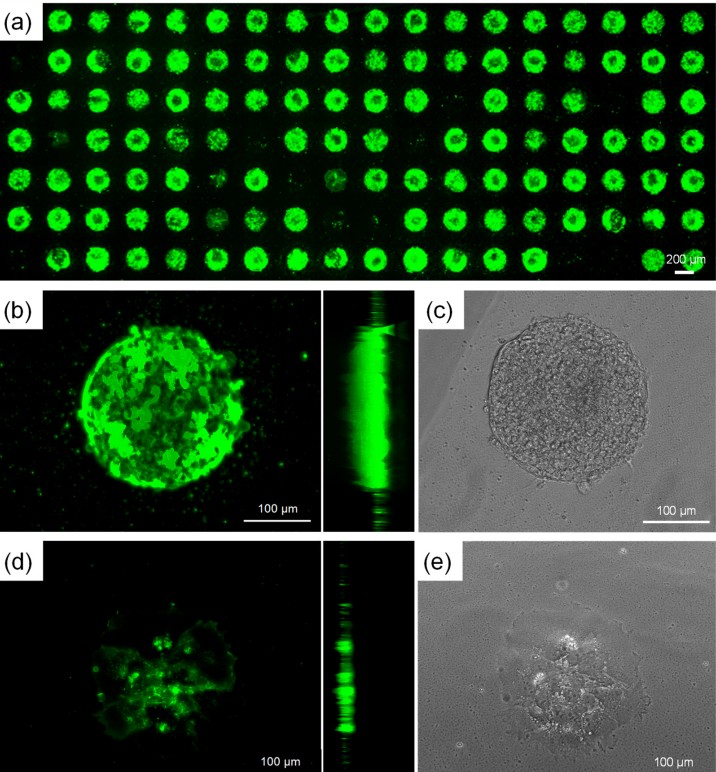

**Figure 5.** (**a**) Fluorescence microscope image of induced pluripotent stem cells (iPSCs) cultured on a cell-nonadhesive hydrophilic polymer patterned substrate for 6 days. Magnified images of stratified (**b**,**c**) and monolayer (**d**,**e**) iPSCs. (**b**,**d**) Fluorescence microscope images with Z-stack measurement. XY (left images) and YZ (right images) planes measured by slicing at 1 μm intervals with the thickness direction as the z-axis. (**c**,**e**) Phase-contrast microscope images.

## 4. Conclusions

In this study, nanostructures formed by colloidal arrays were applied to iPSC cultures. Colloidal arrays were formed using the following procedure. First, core–shell colloids with a polystyrene layer on the surface were adsorbed onto a glass substrate using a layer-by-layer method. Second, the colloids were immobilized via thermal fusion. Third, the surface of the colloids was modified by plasma treatment to improve cell adhesion. By adjusting the number density of colloidal particles immobilized on the substrate, cultured iPSCs were easily detached from the substrate without manual cell scraping, although cell proliferation was reduced compared with flat surfaces. Colloid surface coverage of approximately 15% was optimal for both cell growth and detachment. Furthermore, mesenchymal stem cells, HepG2 cells, umbilical vein endothelial cells, gingival epithelial progenitor cells, and epidermal keratinocytes were tested for growth on colloidal arrays and were found to detach just as easily as iPSCs. These results indicate that colloid array scaffolds are applicable for a variety of cells to enhance cell detachment. In addition to planar culture, cell aggregation of iPSCs attached to the substrate was examined. Using a surface protection film with holes formed by laser processing as a mask, a microscopic pattern of colloidal arrays was formed. Cells were locally aggregated and could be easily detached on these colloidal arrays. Alternatively, by combining hydrophilic polymer patterning of the surface on the colloidal array, multilayered cell aggregates with approximately four layers could be cultured. These cultured aggregates showed iPSC-specific SSEA-4, indicating that iPSCs maintain pluripotency. Furthermore, these multilayered cell aggregates were easily separated and recovered. Although selective detachment to separate iPSCs from other cells has not been achieved, future research may control cell detachment for each cell type because colloidal arrays can regulate cell adhesion. These findings will aid in the development and practicality of applications for controlling the adhesive state and aggregation of iPSCs.

**Supplementary Materials:** The following supporting information can be downloaded at: https://www.mdpi.com/article/10.3390/macromol3020014/s1, Figure S1: Binarized atomic force microscopy images of colloid arrays.

**Author Contributions:** Conceptualization, G.K. and A.M.; methodology, G.K. and A.M.; software, G.K.; validation, G.K.; formal analysis, G.K. and A.M.; investigation, G.K. and A.M.; resources, G.K.; data curation, G.K.; writing—original draft preparation, G.K.; writing—review and editing, A.M.; visualization, G.K.; supervision, A.M.; project administration, A.M. All authors have read and agreed to the published version of the manuscript.

**Funding:** This research received no external funding.

**Data Availability Statement:** Data are available on request.

**Conflicts of Interest:** The authors declare no conflict of interest.

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
