# Peer review of "Easy Cell Detachment and Spheroid Formation of Induced Pluripotent Stem Cells Using Two-Dimensional Colloidal Arrays"

_2673-6209, doi:10.3390/macromol3020014_

Round 1

Reviewer 1 Report

This manuscript describes the study of the influence of nanostructures generated using two-dimensional colloidal arrays on the formation and adhesion of spheroids obtained from induced pluripotent stem cells (iPSCs).
The study of the regulation of cell adhesion of spheroids to the growth support is a very interesting and useful experimental approach for those who work with 3D cell cultures. This paper proposes in an original way the possibility of carrying out the selective detachment of the different cells in culture in the future. The length of this manuscript is suitable, well written and quite interesting.
My observations are: Introduction: Well written and interesting. Materials and Methods: They are well written and a researcher could reproduce the experiments without problems. You should indicate the sterilization method of the colloid system on which the cells were grown. Results and discussion: The results of the experiments are described well. Conclusions: they are a bit short and generic. Try to better explain the importance of the experiments carried out in relation to the topic under discussion. Bibliography: some references are very old, if possible replace them with younger ones. Try to increase the number of updated cited articles. Figures and tables: The figures are clear and easy to read. The captions are well written and report the experimental data well. The manuscript seems to me suitable for the topics covered by the journal. The manuscript can be published with minor changes without problems.

Author Response

Thank you very much for your review of our paper. We have answered each of your points below.

Materials and Methods: You should indicate the sterilization method of the colloid system on which the cells were grown.

Response: We have added the detail of sterilization method in section 2.5.

Conclusions: they are a bit short and generic. Try to better explain the importance of the experiments carried out in relation to the topic under discussion.

Response: We have added the contents of the Conclusions trying to show the importance of our work.

Bibliography: some references are very old, if possible replace them with younger ones. Try to increase the number of updated cited articles.

Response: We have replaced the old reference to new one, increased the number of cited articles.

Reviewer 2 Report

In this study, Kuno et al. studied colloidal arrays for iPSC cultures. By adjusting the density of colloidal particles immobilized on the substrate, cultured iPSCs were easily detached from the substrate without manual cell scraping. However, cell proliferation was reduced compared with flat surfaces. By combining hydrophilic polymer patterning of the surface on the colloidal array, multilayered cell aggregates with approximately 4 layers could be cultured. These multilayered cell aggregates were easily separated and recovered. 

This manuscript looks good. My major concern is because the research did not demonstrate the selective detachment of iPSCs from other cells, whether it is appropriate to say in the title “adhesion control of” stem cells. Besides this, I think this manuscript deserves publication.

Author Response

Thank you for your review of our paper. We have answered each of your points below.

My major concern is because the research did not demonstrate the selective detachment of iPSCs from other cells, whether it is appropriate to say in the title “adhesion control of” stem cells.

Response: We have changed the title as follows.

“Easy cell detachment and spheroid formation of induced pluripotent stem cells using two-dimensional colloidal arrays”